# Synthesizing Remote Sensing and Biophysical Measures to Evaluate Human–wildlife Conflicts: The Case of Wild Boar Crop Raiding in Rural China

**Madeline Giefer [1],* and Li An [2]**

1    Department of Geography, University of North Carolina at Chapel Hill, Chapel Hill, NC 27514, USA
2    Department of Geography and Center for Complex Human-Environment Systems,
     San Diego State University, San Diego, CA 92182, USA; anli@complexities.org
*    Correspondence: mmgiefer@live.unc.edu; Tel.: +1-952-288-7187

**Abstract:** Crop raiding by wild boars is a growing problem worldwide with potentially damaging consequences for rural dwellers' cooperation with conservation policies. Still, limited resources inhibit continuous monitoring, and there is uncertainty about the relationship between the biophysical realities of crop raiding and humans' perceptions and responses. By integrating data from camera traps, remote sensors, and household surveys, this study establishes an empirical model of wild boar population density that can be applied to multiple years to estimate changes in distribution over time. It also correlates historical estimates of boar population distribution with human-reported trends to support the model's validity and assess local perceptions of crop raiding. Although the model proved useful in coniferous and bamboo forests, it is less useful in mixed broadleaf, evergreen broadleaf, and deciduous forests. Results also show alignment between perceptions of crop raiding and actual boar populations, corroborating farmers' perceptions which are increasingly dismissed as a less reliable source of information in human–wildlife conflict research. The modeling techniques demonstrated here may provide conservation practitioners with a cost-effective way to maintain up-to-date estimates of the spatial distribution of wild boar and resultant crop raiding.

**Keywords:** feral pigs; habitat; wildlife management; Landsat; camera trap

## 1. Introduction

Human–wildlife conflict is an increasingly prevalent challenge throughout the world, with potentially severe implications for environmental conservation [1]. These conflicts are centered on competing social and environmental values, and while research cannot point to "correct" solutions, it can help conservation managers predict potential conflicts prior to policy implementation and manage them accordingly. Human–wildlife conflict is a special concern in protected areas and their surroundings. As ecosystems recover, growing populations of some wildlife species can threaten the livelihoods and safety of nearby rural communities [2], especially when these species invade farm fields and devour crops. Not only does crop raiding bring economic hardship to already poor families [3] and contribute to food shortages [4], it can also breed resistance to conservation programs [5] and interfere with their outcomes [3]. Managing wildlife crop raiding is thus socially and environmentally vital. While there have been several studies on crop raiding near protected areas [5–8], few have examined how crop raiding changes over time, likely because field estimates are expensive to obtain [9].

Crucial to the management of human–wildlife conflict is understanding the habitat and distribution of the wildlife species in question. Remotely sensed imagery, especially those from the Landsat satellite series, have long been used to estimate wildlife distribution and habitat suitability due to their cost-free

accessibility [10,11] and their ability to cover large spatial extents [12]. While several studies have simultaneously considered different topographic, hydrologic, and human variables [13], vegetation characteristics remain keystone explanatory variables, usually measured through remotely derived vegetation indices such as the Normalized Difference Vegetation Index (NDVI) [13,14]. In addition, strategies for interpreting results vary, with some studies assigning ordinal suitability ratings ranging from "low" to "very good" [15], while others use a continuous suitability scale [16], and then establish a threshold to determine whether a species is potentially "present" or "absent" at a site [17]. All these models are empirical, requiring on-the-ground population measurements to calibrate the relationships between environmental characteristics and actual wildlife occurrences and distribution. Ground-measurements are increasingly being done using noninvasive camera traps [18,19], although different population distribution indicators have also been employed [11,19].

While many studies that model species distribution using remotely sensed data are undertaken to aid conservation of vulnerable/endangered species [9,20], comparatively few address relationships between economically vulnerable human populations and highly resilient and pervasive wildlife species such as wild boars (*Sus scrofa*). As in many places around the world, forest recovery in China has brought a resurgence of wild boars, especially near protected areas [21], which often has severe economic consequences for farmers, many of whom are poor. In China, this is even more problematic because killing or trapping wild animals is restricted in many places, while alternative methods to control crop raiding, such as fencing, have proven ineffectual [7]. Due to their dense populations, adaptable diets, large bodies, and high reproduction rates, it is difficult to control wild boar populations and/or mitigate their damage to crops [7]. Growth in wild boar populations has even induced cropland abandonment, as is the case in the mountainous Chongqing Province [22], and the problem may be growing in other areas where conservation policies bring increasing forest cover that allows wild boar populations to flourish. This poses a threat to any further conservation action, and data from other regions (e.g., Tianma National Nature Reserve) suggest that these losses offset the social benefits of even voluntary conservation programs [23]. Thus, understanding the distribution of wild boars and crop raiding will improve our ability to assess their burdens to farmers, land-use practices in response to these burdens, and overall long-term dynamics of local ecosystems.

When creating habitat models, it is critical to consider several variables that affect the species' key needs: space, cover, and food. Prior studies have suggested that wild boars prefer deciduous forests in southern Sweden [24] and deciduous or coniferous forests in South Korea [25], although there may be wide regional differences for this extremely widespread species [26]; thus it is necessary to perform site-specific evaluations. Elevation is also a key habitat variable, as lower elevations may provide steadier year-round feeding opportunities, as evidenced by studies in the United States which show that some boars remain at lower elevations while others shift to feed at higher elevations during summer [27]. Meanwhile boars may tolerate steep terrains and in fact be drawn to them when they provide a protective barrier against threats [28]. Furthermore, because slope aspect can influence vegetation through its effects on soil moisture availability [13], aspect values may be reclassified on a scale of 0 to 20 based on Parker's [29] topographic relative moisture index (TRMI). Park and Lee [25] found that east- and southeast-facing slopes were most suitable to wild boars in South Korea, which corresponds to low-to-moderate relative soil moisture values based on the TRMI conversion. Preference for east-facing slopes may also occur because these areas are less likely to be covered by forage-inhibiting snow during the winter months [30], and thus may also exhibit lower soil moistures during spring and summer.

The triadic relationship among farmers, wild boars, and forest growth has been reported by a few studies using empirical data collected within a single season [3,7,31]. This study integrates biophysical and socioeconomic data to illustrate a system in rural China wherein impoverished, small-scale farmers are subjected to an increase in crop losses due to foraging wild boar, a species that stands to flourish amid the region's forest gains, over a period of 26 years from August 1990 to August 2016. Results of this study may provide a cost-effective way to evaluate long-term socioeconomic effects of successful

forest conservation programs, help with the development of appropriate mitigation and compensation programs, and further enhance the long-term success of conservation actions.

## 2. Materials and Methods

### 2.1. Study Site

Fanjingshan National Nature Reserve (N 27°44'42"–28°03'11", W 108°34'19"–108°48'30") is located in northeastern Guizhou Province, southwestern China. Since its establishment in 1978, the reserve has attracted attention from conservationists globally and is inside one of the world's 25 biodiversity hotspots [32]. The mountainous 419-km$^2$ site is dominated by evergreen, deciduous, and mixed broadleaf forests with tracts of bamboo. Elevation varies widely from 400 m to 2560 m. The site is also home to about 13,000 people, most of whom are subsistence farmers, although some have migrated to cities for work or found employment in the area's burgeoning tourism sector [33]. Human settlements in the study area have historically brought instances of both deforestation and reforestation [34]. As in many other mountainous regions inhabited primarily by subsistence farmers, the area has been involved in China's Grain to Green (GTGP) reforestation program since 2000, which aims to provide participants with monetary and/or in-kind payments in return for replacing croplands on steep hillslopes with forestland or grassland [35]. A total of 774 of the area's approximately 3256 households are enrolled in the program, receiving an average of 230 yuan (currently, 1 yuan = 0.14 USD) per mu (15 mu = 1 ha). Through GTGP, the local government has provided farmers with seedlings of pine, Chinese fir, and other tree species, along with bamboo seedlings. Figure 1 contains a true-color aerial image of the reserve in August 2016.

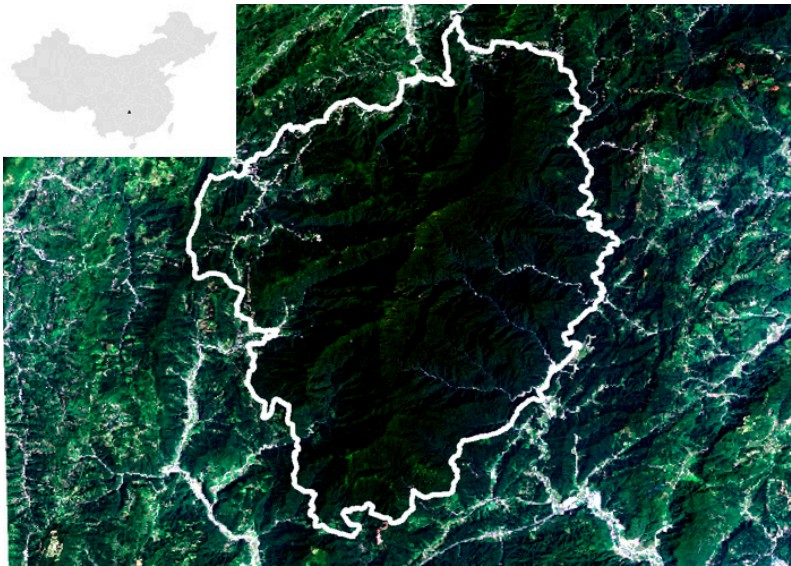

**Figure 1.** True color image of Fanjingshan National Nature Reserve; boundary in white.

### 2.2. Empirical Data

A time series of Landsat imagery taken on near-anniversary dates covering the study area was selected (earthexplorer.usgs.gov; Landsat level-2 product): 22 August 1990, 22 August 1996, 31 August 2002, 16 August 2011 (Landsat 4/5 TM), and 29 August 2016 (Landsat 8 OLI). These dates were chosen because they provided mostly cloud-free near-anniversary dates during summer [36] at intervals of approximately five years [37]. Clouds were identified and masked out across the time series using the cloud/quality layer provided within the Landsat Level-2 product. A relative atmospheric correction was applied to the time series imagery using the empirical line correction [38], pseudo-invariant targets (i.e., cells whose land cover remained invariant across the time series), and the 1990 Landsat TM image

as the base image. To ensure the integrity of the empirical atmospheric correction, pseudo-invariant targets that significantly diverged from the empirical functions were removed until each function had $R^2$ values greater than 0.85. Model coefficients obtained for these empirical atmospheric corrections are available in Appendix A. The resulting equations were then applied to the spectral bands of the other images in the time series. Following this radiometric correction, the wide dynamic range vegetation index (WDRVI) was calculated across each image in the time series $\left(WDRVI = \frac{0.2 \times \rho_{NIR} - \rho_{red}}{0.2 \times \rho_{NIR} + \rho_{red}}\right)$. The WDRVI was chosen because it is sensitive to variation in greenness values when vegetation density is high, whereas other indices saturate [39]. In addition to the Landsat-derived WDRVI, data on elevation were also utilized. These data were derived from a digital elevation model obtained by the Shuttle Radar Topography Mission (earthexplorer.usgs.gov), from which slope and slope aspect were calculated. Elevation, slope, and slope aspect were assumed constant throughout the time series.

Data on wild boar populations were obtained using 69 camera traps (Bushnell Trophy Cam) deployed between April 2015 and November 2016 throughout the study area [40]. Cameras were placed in each of 71 sampling plots (20 m × 20 m), which had been selected to spread across most of the reserve area. In total, 69 of these camera trap placements produced usable observations. While accessibility issues inhibited true random placement, sampling plots were selected based on distance to other plots, elevation, and the advice of reserve staff and local guides. To fit with the anniversary dates of the Landsat time series, which are ideally collected in summer due to the season's phenological stability [41], only camera trap observations from summer were used (1 June–31 August). Boar density at each camera trap location was estimated by dividing the total number of boars sighted by the number of summer days that camera was in operation. To prevent the same boar from being counted more than once in rapid succession, it was assumed that a spurt of snapshots captured within 10 min were of the same boar unless more than one appeared in a single image. Where there was more than one per image, the number of boars was assumed to be the greatest number captured in one image during that cluster. A continuous estimate of this proxy for population density was then derived across the entire study site using the inverse distance weighting interpolation (IDW) procedure, using three different *k*-values: 1, 2, and 3. This set of *k*-values is typical in environmental research [42] and allowed us to optimize model calibration by selecting the power that produced the most transferrable results. This produced three separate surfaces, each estimating boar population density across the landscape; *k*-values refer to the power assigned to each point's distance from the cell in question; when $k = 1$, the weight would be proportional to the inverse distance; when $k = 2$, the weight would be proportional to the inverse distance-squared, and so on. Thus, the surface derived with $k = 1$ gave the most weight to camera traps farther away and those with $k = 3$ gave the least. Figure 2 illustrates the estimated boar density distribution produced by each *k*-value.

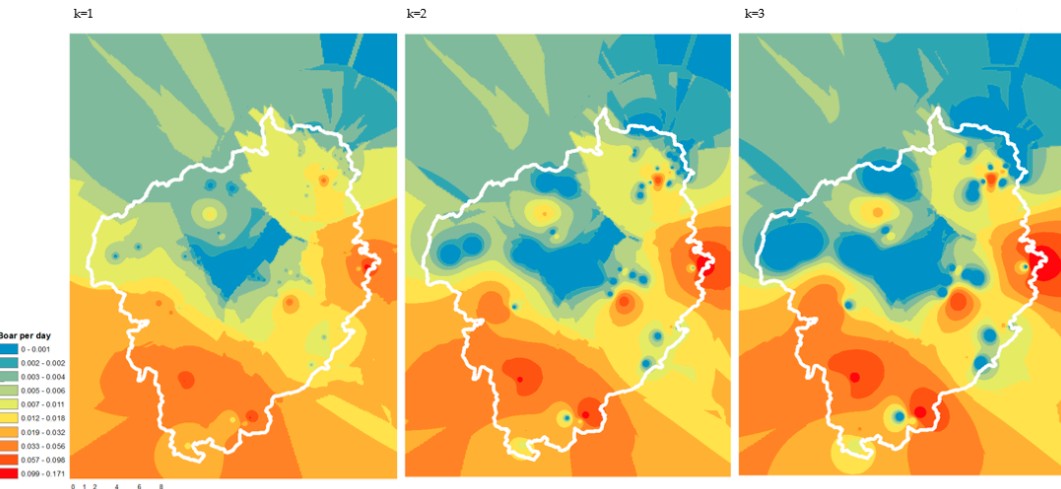

**Figure 2.** IDW interpolated boar density by *k*-value.

Each camera trap location was also assigned to one of five vegetation classes recorded based on in-situ observation: evergreen broadleaf, bamboo, coniferous, mixed broadleaf, and deciduous. The camera trap locations were converted to Thiessen polygons to estimate vegetation cover across the study area. Camera trap coordinates, dates of operation, and vegetation classes are listed in Appendix B. From here, the estimated boar density and vegetation type were assigned to each of 1527 agricultural plots represented as points—73 of which were removed from analysis because they fell outside the range of the Thiessen polygons and could not be reliably assigned to a forest type. In addition, 156 parcels were removed due to missing household allocations. This left 1298 points (i.e., parcels) in the model. Each point was also assigned the zonal mean WDRVI, elevation, slope, and slope aspect within a 500 m radius; 500 m was chosen to reflect the typical 500–600 m home range of a wild boar [43]. Figure 3 illustrates vegetation distribution and camera trap and parcel locations, with the reserve boundary in white.

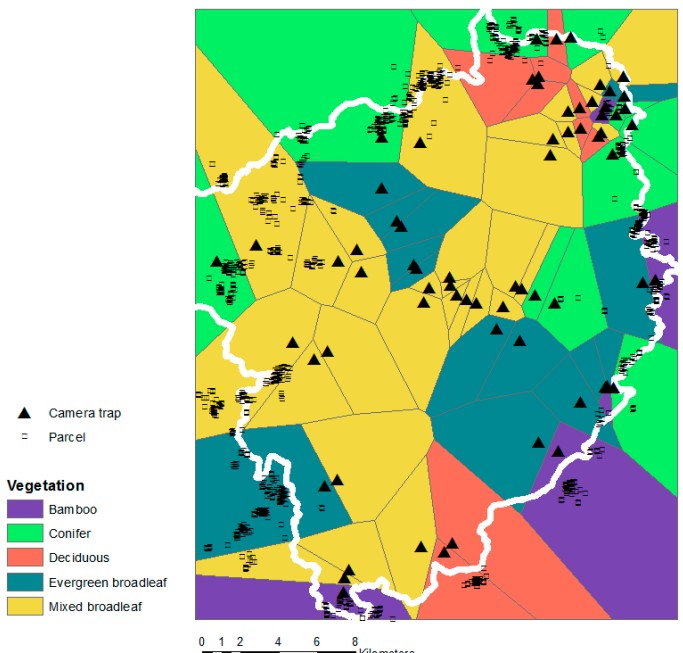

**Figure 3.** Map of study site, camera traps, parcels, and vegetation types.

In-person interviews were conducted with households in a stratified random sample. Based on a 2013 census that had identified 3256 households, 1160 households were selected in hopes of obtaining 650 usable interviews after eliminating households with no knowledgeable member present due to travel or other circumstances. The target of 650 households was selected somewhat arbitrarily to ensure satisfactory statistical power. The 3256 households were divided into 123 sampling units; 58 of these units were selected and assigned to 20 administrative villages in proportion to each village's population size. This produced a slight overrepresentation of smaller villages. In total, 20 households were selected from each administrative village per sampling unit; 1160 households were ultimately selected, and full surveys were completed for 605 households in 2014. In 2015, these 605 households were revisited, and 494 full surveys were completed [44]. Details are available at http://complexities.org/pes/research/recent-updates. The household head was interviewed when possible; otherwise any other available, knowledgeable adult was interviewed. Each interviewee indicated the level of crop raiding over the past twelve months on a five-point Likert scale where 1 = "no crop raiding," 2 = "non-serious crop raiding," 3 = "somewhat serious crop raiding," 4 = "serious crop raiding," and 5 = "very serious crop raiding." The Likert scale is useful for analyzing ordinal data like this when the distances between values, i.e., perceived seriousness, cannot be practically measured [45]. Respondents also answered whether the severity had increased, decreased, or remained constant over

the past 10 years. These interviews also covered each household's demographics, work and migration history, participation in GTGP, experience with crop raiding, agricultural holdings, and other topics related to livelihood and lifestyle. This produced a total of 605 household interviews; in 2015, all 605 households were revisited to provide more information on land use and assist in participatory mapping of their agricultural holdings. A total of 494 household interviews were completed in 2015. Each household identified the location of its agricultural parcels on a local map, which were later converted to point coordinates. This provided a dataset with 1298 agricultural fields, each assigned to a household with the household's respective survey responses.

### 2.3. Data Analyses

To evaluate the ability to predict boar density from WDRVI, slope, elevation, and aspect, separate Ordinary Least-Square (OLS) regression analyses were run for each vegetation type and for all five vegetation types combined, using the three interpolated boar densities (obtained using three different $k$-values in the IDW) as dependent variables. Prior studies have shown OLS regressions are useful for habitat suitability models [46] despite imperfectly linear relationships and may even fit the data better than alternatives like nonlinear quantile regression [47]. While a nonlinear function like a general additive model may have provided slightly more accurate predictions of boar density at the validation parcels based on the calibration parcels, because the OLS stage of this analysis was simply meant to test the usefulness of these variables at each vegetation type before geographically weighted analysis later on, we opted for the simplicity of OLS. Separate regressions for each vegetation type were run to account for differences in how attracted boar are to different types [24]. These OLS models were calibrated using clustered random samples of agricultural fields for each vegetation class; parcels were chosen via a random number generator where the parcel with the unique ID matching that number, and those with the next four unique IDs, were selected. This clustering technique was selected to save time over simple random sampling of one parcel at a time, while the clusters were small enough (5 out of 1298 parcels at a time) to help ensure wide distribution and relative independence among selected parcels. This selection process was repeated until 25% of the parcels had been chosen; these would become the "validation" group and the rest would form the "calibration" group. The linear models derived from each calibration group were then applied to the validation parcels within the same vegetation class, and a new OLS regression was run to relate predicted to observed (i.e., interpolated) boar density. A positive, statistically significant relationship between predicted and observed values indicated that the model was useful in predicting boar density. Closer slope estimates to 1.0, together with intercepts closer to zero, showed that the model exhibited higher accuracy and transferability. After calibrating and validating models for each $k$-value and each vegetation type, we needed to select the $k$-value that best reflected the dispersal of wild boar around the camera traps. The best $k$-value is that for which predicted and measured values are the closest, which can be answered using $R^2$ [48]. Regressions based on the $k = 1$ interpolated boar density produced the highest $R^2$ values in calibration for three of the five forest types and the combined model. It also provided the highest $R^2$ values when validating the regressions for four of the five forest types and the combined model. Thus, the IDW interpolation using $k = 1$ was used for the remainder of analysis. Models obtained using the calibration and validation datasets are listed in Table 1.

**Table 1.** OLS regression models for interpolated boar population density in 2016.

| Vegetation Type | Linear Regression | n | R$^2$ |
|---|---|---|---|
| Evergreen broad | $BPD = 0.034 + 0.007(WDRVI) + 0.0004\,^c(Slope) - 0.000004(Elev) - 0.0008\,^c(TRMI)$ | 189 | 0.18 |
| | $BPD = -0.008 + 1.226\,^c(Predicted)$ | 63 | 0.19 |
| Bamboo | $BPD = 0.025 - 0.001(WDRVI) + 0.003\,^c(Slope) - 0.00008\,^c(Elev) - 0.0004(TRMI)$ | 161 | 0.49 |
| | $BPD = -0.009 + 1.460\,^c(Predicted)$ | 53 | 0.51 |
| Conifer | $BPD = 0.028 + 0.014\,^c(WDRVI) + 0.0001\,^b(Slope) - 0.00003\,^c(Elev) - 0.0002\,^c(TRMI)$ | 283 | 0.68 |
| | $BPD = 0.0007 + 0.974\,^c(Predicted)$ | 95 | 0.55 |
| Mixed broad | $BPD = 0.057 + 0.061\,^c(WDRVI) - 0.0002(Slope) - 0.00005\,^c(Elev) - 0.00003(TRMI)$ | 295 | 0.21 |
| | $BPD = -0.010 + 1.518\,^c(Predicted)$ | 98 | 0.38 |
| Deciduous | $BPD = 0.008 + 0.066\,^c(WDRVI) + 0.003(Slope) - 0.00007\,^c(Elev) - 0.00004(TRMI)$ | 46 | 0.98 |
| | $BPD = 0.041 - 0.049\,^a(Predicted)$ | 15 | 0.19 |
| Combined | $BPD = 0.053 + 0.048\,^c(WDRVI) + 0.0005\,^c(Slope) - 0.00004\,^c(Elev) - 0.0010\,^c(TRMI)$ | 974 | 0.45 |
| | $BPD = -0.0009 + 1.086\,^c(Predicted)$ | 324 | 0.43 |

*a*: $p < 0.10$, *b*: $p < 0.05$ and *c*: $p < 0.01$ (based on robust standard errors).

After investigating whether remotely sensed variables could be used to predict wild boar density, a new series of regression analyses was run to predict boar population density over time. First, the same variables from the OLS models were put into a geographically weighted regression (GWR) model to estimate a spatially variable model of boar density based on vegetation type, WDRVI, elevation, slope, and aspect-derived moisture index in the 500 m surrounding each agricultural field. GWR was selected as it is likely to provide more spatially accurate estimations of the effects of environmental variables than global OLS models could when spatial variability in the research question is of importance [49]. (Slope was not included as a coefficient for deciduous forests because its values were too clustered for the software to process without error; we deemed this an acceptable omission given slope's lack of significance in the OLS model for deciduous forests.) These models were calibrated using the WDRVI values from the 2016 Landsat image and the slope, aspect, and elevation values from the DEM, then the point-specific coefficients were applied to the 1990, 1996, 2002, and 2011 Landsat images to estimate boar exposure on each field during those years. The model was applied across multiple years to estimate whether boar populations had increased or decreased overall over time and whether our modeled population densities correlated with crop raiding reported by farmers. These years were selected because they provided mostly cloud-free near-anniversary dates and covered periods before and after the implementation of GTGP in 2000. The boar density value at each parcel for each year was then subtracted from the GWR-predicted value based on the 2016 image, producing four new variables estimating change in boar density between the given year and 2016. This was an appropriate way to estimate historical boar distributions because habitat suitability, which can be modeled through indicators like vegetation index and topography, is a widely used indicator of population distribution [12,16]. Figure 4 illustrates the methodology for estimating boar density in current and past years. This dataset was then joined by household ID with the 2015 household survey dataset ($n = 494$). These records went into a random effects logistic regression grouped by household, where the dependent variable was whether or not the household had experienced "serious" or "very serious" crop raiding over the past 12 months (scores 4–5 on the Likert scale), and the independent variable was the GWR-predicted boar density from the 2016 image. A separate regression was run for each vegetation type. A similar set of regressions was then run wherein the dependent variable was whether the respondent believed crop raiding had decreased in the past 10 years and the independent variable was the estimated change in boar density at the parcel since 1990, 1996, 2002, and 2011. This was to help validate our model of historical boar population distribution by comparing it with the local understandings of people who would have experienced these fluctuations firsthand. With five vegetation types and four images prior to 2016, this produced an additional 20 regressions.

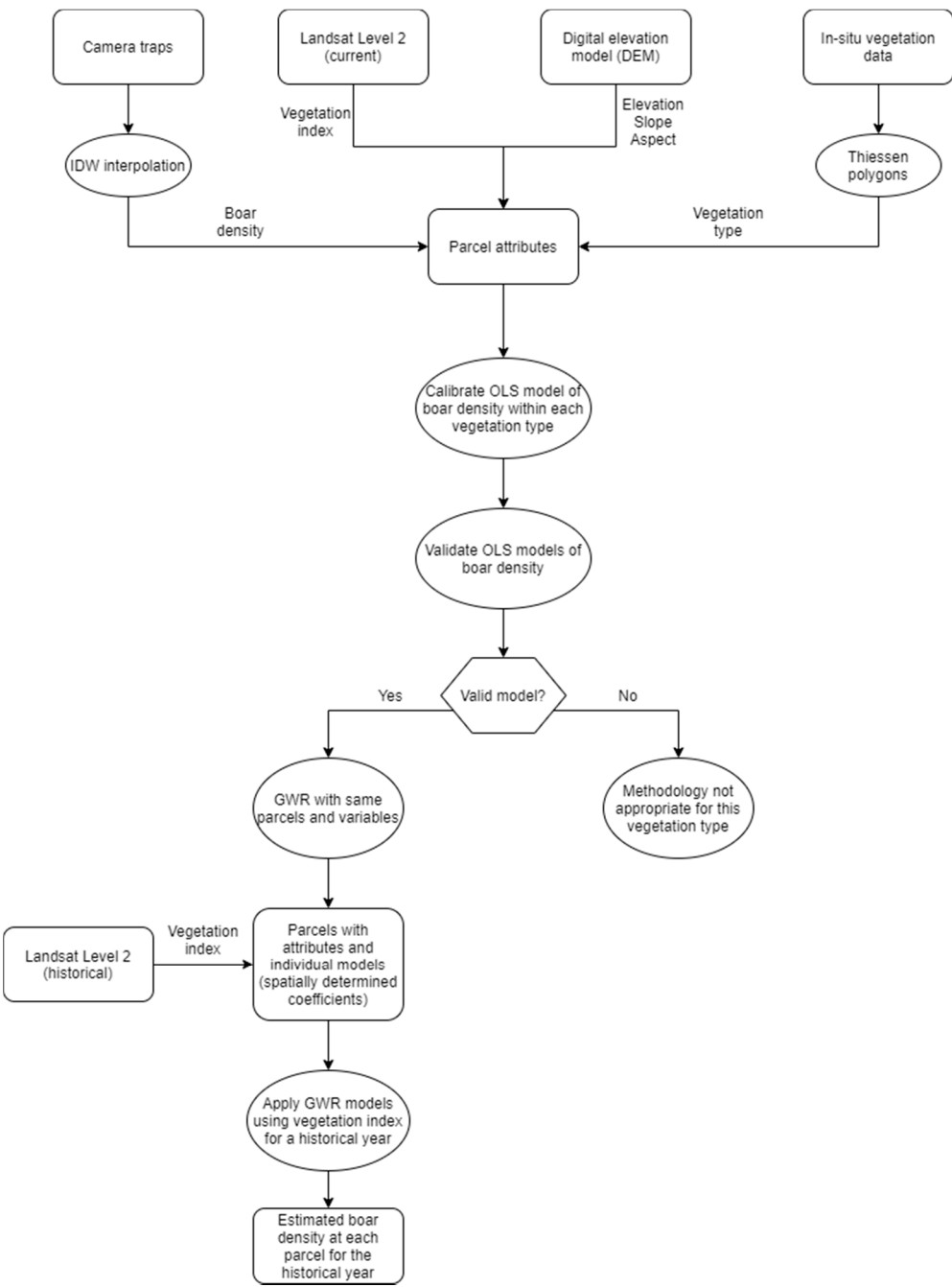

**Figure 4.** Methodological framework for estimating boar density across years.

## 3. Results

Table 2 shows the mean estimated boar density for each vegetation type. Figure 5 shows the percentage of 446 households that rated crop raiding at each level of seriousness based on the Likert scale from "none" to "very serious," as well as the percentages who believed crop raiding had lessened, remained steady, or increased over the past 10 years. 40.2% believed crop raiding had worsened in the past 10 years, 21.4% believed it had remained steady, and 28.4% believed it had lessened. Figure 6 illustrates how these responses were distributed across the reserve. We used a survey conducted on the same sample the previous year (2014) to confirm that wild boars were the most-blamed species for crop raiding, with 93.7% of respondents who experienced crop raiding stating that wild boar was the species that caused the most damage. This confirmed boar population was an appropriate proxy

for crop damage risk. "Boar per day" estimates for each camera trap ranged from 0 to 0.18; 48 of the 71 camera traps did not record any boar during summer, and the highest number of boars observed in one day was four.

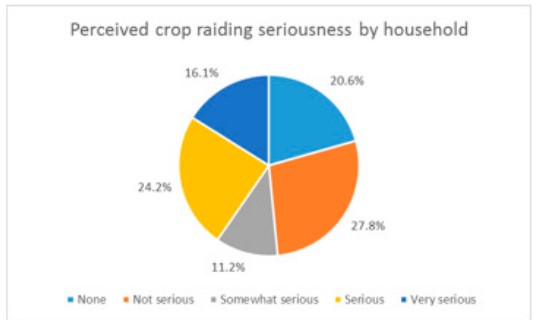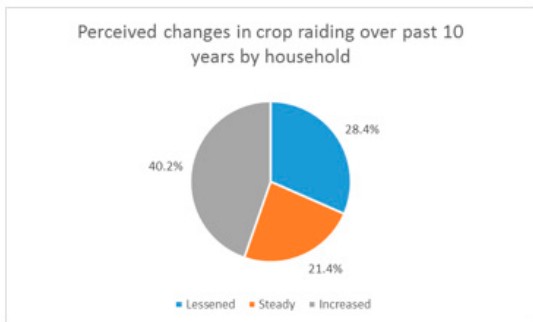

**Figure 5.** Perceived seriousness and trajectory of crop raiding.

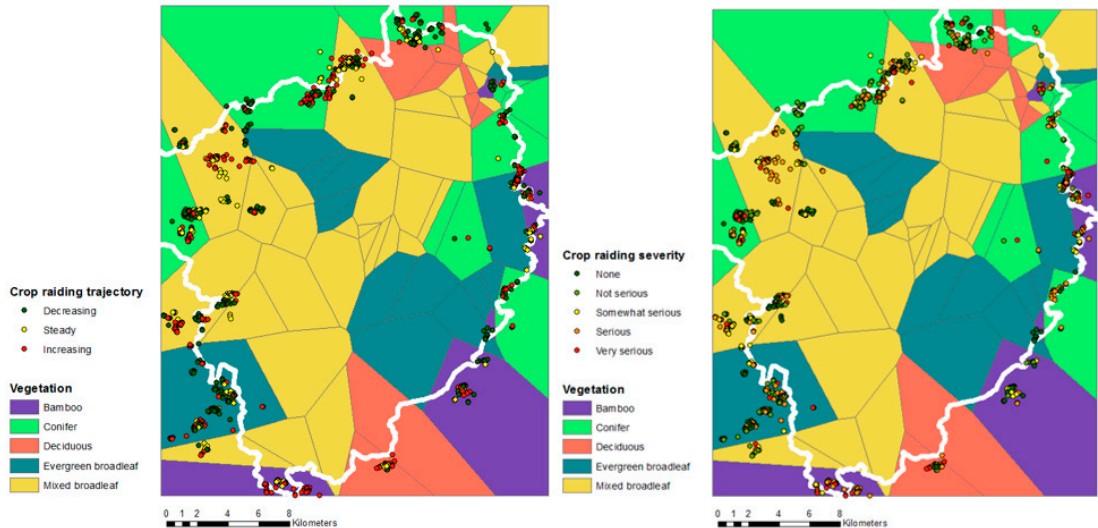

**Figure 6.** Household perceptions of crop raiding throughout the region.

On average, estimated boar density decreased slightly within each 500 m radius around agricultural fields between 1990 and 2016. Across all vegetation types, predicted boar density decreased by a mean of 0.00005 BPD (SD 0.0002). However, trends varied widely by vegetation type. Mean change in boar density by agricultural field since 1990, 1996, 2002, and 2011, along with the maximum increase and decrease, are listed by vegetation type in Table 3. Predictive model accuracies also varied by vegetation type. When a single model was applied to all forest types at once, it performed respectably ($R^2 = 0.45$) and generally followed the correlations found in separate models. Across the entire study area, boar density was positively explained by vegetation index (+0.048 BPD per unit WDRVI; $p < 0.01$), positively explained by slope (+0.0005 BPD per degree; $p < 0.01$), negatively explained by elevation (−0.004 BPD per 100 m; $p < 0.01$), and negatively explained by aspect-derived moisture index (−0.0010 BPD per point; $p < 0.01$). This suggests boars overall prefer lower, dryer, steeper slopes with denser vegetation. The model was strongest in conifer forests with high internal consistency (i.e., accuracy of the model within the spatial range of calibration sites) at ($R^2 = 0.68$) and was well validated ($\beta_1 = 0.974$; $\beta_0 = 0.0007$; $p < 0.01$; $R^2 = 0.55$). In conifer forests, boar density was positively correlated with vegetation index (+0.014 BPD per one-unit increase in WDRVI; $p < 0.01$), positively correlated with slope (+0.0001 BPD per degree; $p < 0.05$), negatively correlated with elevation (−0.003 BPD per 100 m; $p < 0.01$); and negatively correlated with aspect-derived moisture index (−0.0002 BPD per point; $p < 0.01$). The bamboo model was moderately strong; there was moderate internal consistency

within the model ($R^2$ = 0.49) and it fit the validation parcels well ($\beta_1$ = 1.460; $\beta_0$ = −0.009; $p$ < 0.01; $R^2$ = 0.51). In bamboo forests, boar density was positively correlated with slope (+0.003 BPD per degree; $p$ < 0.01), negatively correlated with elevation (−0.008 BPD per 100 m; $p$ < 0.01), and uncorrelated with vegetation index or aspect. In evergreen broadleaf domains, where the model had a somewhat low internal consistency ($R^2$ = 0.18), boar density was positively correlated with slope (+0.0004 BPD per degree; $p$ < 0.01), negatively correlated with aspect-derived moisture index (−0.0008 BPD per point; $p$ < 0.01), and uncorrelated with vegetation index and elevation. The validation performed similarly well to the calibration ($\beta_1$ = 1.226; $\beta_0$ = −0.008; $p$ < 0.01; $R^2$ = 0.19). Model $R^2$ for mixed broadleaf forests was also moderately low ($R^2$ = 0.21), and the model applied less effectively to validation parcels ($\beta_1$ = 1.518; $\beta_0$ = −0.010; $p$ < 0.01; $R^2$ = 0.38). Here, boar density was positively correlated with vegetation index (+0.061 BPD per unit WDRVI; $p$ < 0.01), negatively correlated with elevation (−0.005 BPD per 100 m), and uncorrelated with slope and aspect. The deciduous forest model had the highest internal consistency; with an $R^2$ of 0.98, it showed boar density positively correlated with vegetation index (+0.066 BPD per unit WDRVI; $p$ < 0.01), negatively correlated with elevation (−0.007 BPD per 100 m; $p$ < 0.01), and uncorrelated with slope or aspect. Despite the high internal consistency, model validation failed; the relationship between predicted and interpolated boar density was negative. This was unsurprising given the smaller sample size and dense clustering of most deciduous plots. Overall, estimated boar density decreased slightly across the study site between 1990 and 2016; it decreased in evergreen broadleaf, conifer, and mixed broadleaf forests, but increased in bamboo and deciduous domains. Still, the direction and magnitude of change varied considerably within each vegetation type.

**Table 2.** Mean estimated boar density by vegetation type.

| Vegetation Type | Est. Boar Per Day | One Boar Per—Days |
|---|---|---|
| Evergreen broad | 0.034 (SD = 0.007) | 30 |
| Bamboo | 0.029 (SD = 0.017) | 35 |
| Conifer | 0.007 (SD = 0.005) | 133 |
| Mixed broad | 0.014 (SD = 0.012) | 72 |
| Deciduous | 0.032 (SD = 0.013) | 31 |
| Combined | 0.019 (SD = 0.015) | 52 |

**Table 3.** Mean change in estimated boar per day between years, and maximum increase and maximum decrease between 1990 and 2016.

| Vegetation Type | Mean Change 1990–2016 | Mean Change 1996–2016 | Mean Change 2002–2016 | Mean Change 2011–2016 | Max. Increase 1990–2016 | Max. Decrease 1990–2016 |
|---|---|---|---|---|---|---|
| Evergreen broad | −0.00007 | −0.0005 | −0.0017 | +0.0005 | +0.00006 | −0.0013 |
| Bamboo | +0.0014 | -0.00004 | +0.0008 | +0.0002 | +0.0134 | −0.0029 |
| Conifer | −0.0001 | −0.0002 | −0.0003 | +0.0001 | +0.0014 | −0.0023 |
| Mixed broad | −0.0009 | −0.0013 | −0.0021 | +0.0005 | +0.0063 | −0.0114 |
| Deciduous | +0.0007 | +0.0049 | +0.0014 | +0.0019 | +0.0082 | −0.0018 |
| Combined | −0.00005 | −0.0002 | −0.0006 | +0.0003 | +0.0134 | −0.0114 |

Geographically weighted regressions reflected survey-reported perceptions of crop raiding in some vegetation types. In conifer and deciduous forests, higher estimated boar density correlated with a greater likelihood that a householder had experienced "serious" or "very serious" crop raiding over the previous twelve months. Relationships between predicted boar density and reported crop raiding severity were not statistically significant for evergreen broadleaf, bamboo, or deciduous forests. Probability of reporting crop raiding as "serious" or "very serious" is illustrated in Figure 7. Changes in estimated boar density also correlated with reported changes in severity for some years in evergreen broadleaf, conifer, and deciduous forests. In evergreen broadleaf forests, respondents who

had experienced greater decreases in boar density since 1990 or 1996 were more likely to report crop raiding having decreased during the past 10 years. This was also the case for conifer and deciduous forests that had experienced greater decreases in estimated boar density since 1996. Estimated changes in boar density did not correlate with the likelihood of reporting lessened crop raiding in bamboo or mixed broadleaf forests for any year. Figure 8 illustrates the probability a respondent reported crop raiding as decreasing if predicted boar density had decreased by 0.001 BPD between each comparison year and 2016. All logistic regression models are listed in Table 4.

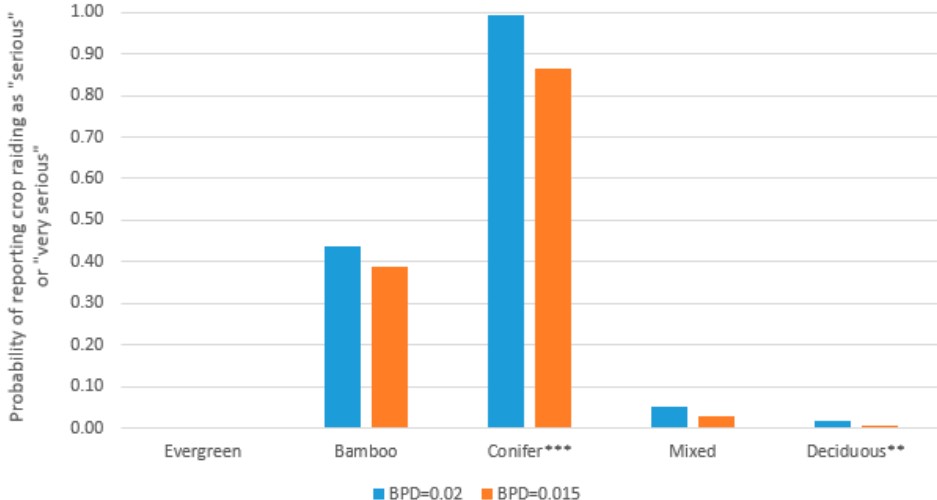

**Figure 7.** Probability of reporting crop raiding as "serious" or "very serious" by vegetation and two plausible boar densities (* $p < 0.05$; ** $p < 0.01$).

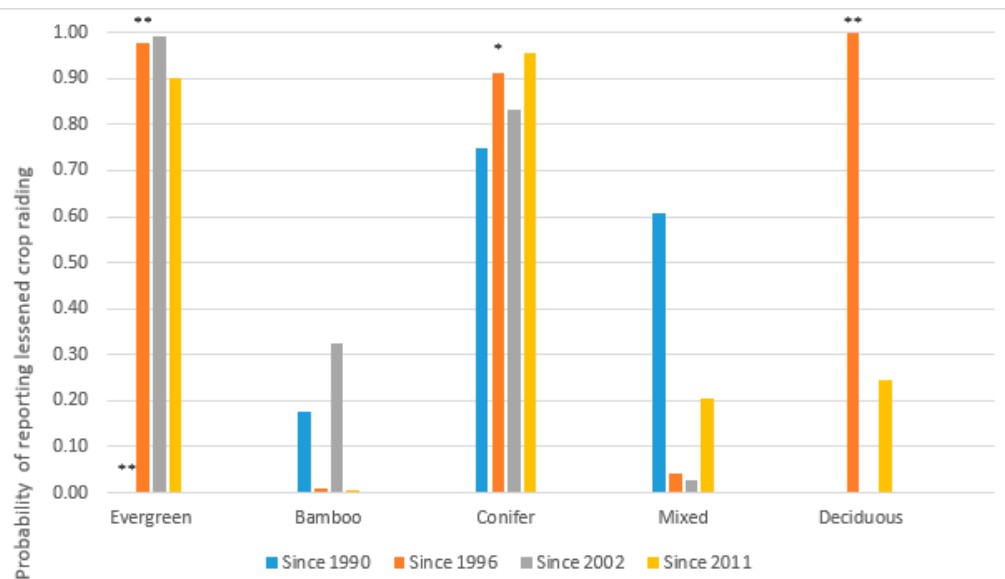

**Figure 8.** Probability of reporting crop raiding having lessened in the past 10 years if estimated boar density decreased by 0.001 BPD between the given year and 2015 (* $p < 0.05$; ** $p < 0.01$).

**Table 4.** Log-odds of reporting crop raiding as "serious" or "very serious" in the 2015 survey based on interpolated boar density ("In 2015"); log-odds of reporting crop raiding as becoming less severe in the past 10 years based on estimated changes in boar density between a given year and 2016 ("Since [year]").

| Vegetation | In 2015 | Since 1990 | Since 1996 | Since 2002 | Since 2011 |
|---|---|---|---|---|---|
| Evergreen | $-22.6 - 340(d)$ | $-1.73 - 9967(\Delta d)$ ** | $-2.21 - 6025(\Delta d)$ ** | $3.62 - 1324(\Delta d)$ | $-3.73 - 597(\Delta d)$ |
| Bamboo | $-1.10 + 42.6(d)$ | $1.28 - 259(\Delta d)$ | $-3.75 + 1030(\Delta d)$ | $0.807 - 78.9(\Delta d)$ | $-4.69 + 625(\Delta d)$ |
| Conifer | $-7.85 + 646(d)$ ** | $-0.502 - 1596(\Delta d)$ | $-2.05 - 4395(\Delta d)$ * | $-1.10 - 2689(\Delta d)$ | $1.18 - 1860(\Delta d)$ |
| Mixed | $-5.15 + 111(d)$ | $0.416 - 15.0(\Delta d)$ | $-3.54 - 411(\Delta d)$ | $-4.07 - 470(\Delta d)$ | $-1.41 - 52.2(\Delta d)$ |
| Deciduous | $-8.30 + 217(d)$ ** | $6.07 - 488(\Delta d)$ | $31.8 - 13233(\Delta d)$ ** | $-18.9 + 523(\Delta d)$ | $-1.17 - 43.3(\Delta d)$ |

d = predicted boar density (boar per day) from 2016 image and GWR regressions; * $p < 0.05$; ** $p < 0.01$.

## 4. Discussion

Our results suggest remotely sensed variables are useful for estimating boar population density and crop raiding severity under certain vegetation types. Bamboo and coniferous forests produced models with moderate to high internal consistency and transferability (moderately high $R^2$ in the OLS calibration and validation models within each forest type). In these vegetation types, remotely sensed vegetation indices and topography seem to be useful for estimating boar density with minimal costs, and it is also feasible to estimate year-to-year variations in wild boar population density. However, this should be further verified with multiyear ground estimates of boar populations. Although the modeling strategy was less internally consistent and transferrable for mixed and evergreen broadleaf forests than it was for bamboo and coniferous forests, it still may provide some useful information, but results from this model need to be interpreted more cautiously. Furthermore, the model's lack of significant relationship with vegetation index in evergreen broadleaf and bamboo domains suggests this technique would not be useful in providing multi-year estimates there because vegetation index is the only predictive variable that can change substantially from year to year. The model for deciduous forests, although exhibiting extremely high internal consistency, exhibited no transferability as the relationship between predicted and interpolated values was negative and small. This contradiction may have occurred due to the smaller sample size and densely clustered pattern of the deciduous units; most of these fields were densely packed at the south end of the reserve with a few fields scattered along the northern end. The randomly selected calibration parcels all happened to occur at the south end, so it is unsurprising that the model would have poor predictive ability in parcels located more than 30 km away and whose vicinity was not represented in the calibration. With better dispersed survey units, it may be feasible to derive a usable model for deciduous forests using this method. It should also be noted that camera traps were generally placed in more geographically accessible locations due to the region's rough terrain [40]; this may have limited our model's usability in some regions.

While this dataset does not allow us to verify crop damage or historical boar populations through ground estimates, household survey responses offer insights that can be used to corroborate the models. In conifer and deciduous forests, households whose fields showed higher risk of crop raiding by wild boars were more likely to report having experienced "serious" or "very serious" crop raiding in the past 12 months. This suggests that the model constitutes a useful predictor of boar density in these forest types. In conifer, evergreen broadleaf, and deciduous forests, households whose fields had higher estimated decreases in boar exposure since 1996 were more likely to report that crop raiding had decreased in the past 10 years; this was also true for changes since 1990 in evergreen broadleaf forests. While there was no correlation between the estimated boar densities since 2002 or 2011 and the reported crop raiding trajectory, this does not necessarily undermine the hypothesized relationship during the decade in question. It cannot be assumed that respondents confined their thought processes to the 10 years specified in the survey (2005–2015), as it is common for respondents to "telescope" prior years into a given period, especially regarding routine, non-landmark events [50]. Their responses may thus reflect changes prior to the decade defined by the survey if changes before 2005 are "telescoped" in. Further, although our results suggest our modeling framework best captures population conditions

in conifer forests, it is important to underline that these forests have the lowest boar densities among the vegetation types evaluated despite conifer forests' wide geographic coverage and high sample size ($n$ = 378). While this may reduce the model's utility at this study site, it shows our methods are more useful in conifer-dominated landscapes that experience substantial crop raiding by wild boars.

The results obtained in this study address a critical issue in human–wildlife conflict: that of gaps between human perceptions of wildlife actions and biophysical realities [51]. Human–wildlife conflict is fraught with misconceptions, and farmers often overestimate the threat that wildlife pose to their crops [52]. This can exacerbate the resistance to conservation efforts and environmental gains [3,5]; it may also lead to misdirected efforts to mitigate farmers' losses and reduce tensions. However, this study's correlations between perceived crop raiding occurrence, and its temporal trajectories, with modeled boar population density provides preliminary evidence that perceptions and reality are often in sync, at least within our study area. Under these circumstances, technical assistance to reduce crop raiding may effectively reduce tensions and preserve cooperation with conservation actions. Of course, given that this dataset does not measure actual crop damage or ask landholders for more detailed information on their experiences, this study does not provide sufficient proof on its own. Further research with objective, physical measurements of crop damage, together with more detailed information on landholders' perceptions, are needed to verify correlations among modeled boar density, actual crop raiding, and perceived crop raiding. Nevertheless, this study does provide nascent support for the validity of crop raiding perceptions as well as a methodological skeleton for further investigation with more detailed data.

The authors emphasize this study is exploratory, intended more to test the feasibility of the methodological framework than to provide concrete answers on wild boar and crop raiding at Fanjingshan. Along these lines, it is important to note that forecasting based on population estimates from such a limited timeframe is dubious; this methodology would be better employed with camera trap data from several years, ideally spread out. Adding temporal depth to the camera trapping estimates would also allow for the inclusion of critical climatic data like year-to-year variations in temperature and moisture, which may vastly improve habitat and population predictions. This study indeed demonstrates the utility of camera traps, remotely sensed images, and household surveys in deriving empirical relationships between boar distributions and present-day crop raiding burdens. Meanwhile the correlations between estimated boar distributions over time and landholders' perceptions of worsening or abating crop raiding suggest the modeling techniques described here may be useful for updating habitat maps and planning interventions based on time-variant, remotely sensed variables like vegetation and weather. Creating reliable forecasts of boar population and crop damage using these methods will, of course, call for richer data from which to calibrate the models. Future studies should refine this methodology with multi-year camera trap measurements (or other reliable population estimation techniques), weather data, and physical measurements of crop damage that do not rely solely on landholders' perceptions. After investing in a robust foundation of data, this modeling technique may provide an effective, affordable means of managing wildlife and their damage to crops as populations fluctuate and shift over time.

## 5. Conclusions

This study uncovers the potential to improve crop raiding monitoring and management over multi-decade periods at minimal cost after initial ground population estimates are made. This may help design cost-effective, easily updatable compensation schemes and technical interventions to minimize economic burdens to farmers. It also empirically affirms anecdotal understandings of crop raiding's relationship to regional ecological changes in some vegetation domains. Although the model suggests crop raiding is not increasing for all households, it demonstrates fine-scale heterogeneity in both realities and perceptions. While the boar density models obtained for Fanjingshan cannot (and should not) be applied to other areas directly [13], the methods for deriving geographically weighted coefficients described here may allow for cost-effective, long-term monitoring of wild boar

populations and crop raiding risks at other sites around the world. Given the expense of continuous ground measurements and the limited resources in many affected communities, this may improve boar management and compensation arrangements at minimal cost by allowing practitioners to update distribution maps using remotely sensed imagery.

**Author Contributions:** Conceptualization, M.G.; Methodology, M.G.; Formal Analysis, M.G.; Data Curation, L.A.; Writing-Original Draft Preparation, M.G.; Writing-Review & Editing, L.A.; Funding Acquisition, L.A. All authors have read and agreed to the published version of the manuscript.

**Funding:** This research was funded by the National Science Foundation under the Dynamics of Coupled Natural and Human Systems program, grant number DEB 1212183 and BCS-1826839.

**Acknowledgments:** We gratefully acknowledge the data collection efforts of Shuang Yang, Cindy Tsai, Stuart C. Aitken, Douglas A. Stow, Rebecca Lewison, Richard E. Bilsborrow, Minjuan Wang, and Xiaodong Chen. Additional thanks to Andrés Viña for his guidance on image processing and statistical analysis. This work was made possible by the cooperation of the staff and residents of Fanjingshan National Nature Reserve.

**Conflicts of Interest:** The authors declare no conflict of interest.

## Appendix A

**Table A1.** Empirical line corrections for radiometric rectification.

| Band | Year | Equation | $R^2$ | $n$ |
|------|------|----------|-------|-----|
| NIR | 1996 | $R_{Corr} = 0.831 \times R_{1996} + 534.56$ | 0.85 | 258 |
| | 2002 | $R_{Corr} = 0.952 \times R_{2002} + 342.27$ | 0.85 | 258 |
| | 2011 | $R_{Corr} = 1.022 \times R_{2011} - 47.223$ | 0.86 | 249 |
| | 2016 | $R_{Corr} = 0.906 \times R_{2016} + 218.48$ | 0.85 | 234 |
| Red | 1996 | $R_{Corr} = 1.226 \times R_{1996} + 20.32$ | 0.91 | 235 |
| | 2002 | $R_{Corr} = 1.113 \times R_{2002} + 69.99$ | 0.86 | 248 |
| | 2011 | $R_{Corr} = 0.892 \times R_{2011} + 99.455$ | 0.85 | 243 |
| | 2016 | $R_{Corr} = 1.034 \times R_{2016} + 138.25$ | 0.85 | 240 |

## Appendix B

**Table A2.** Camera trap coordinates and dates of operation.

| Plot ID | Start Date | End Date | Long. (WGS84) | Lat. (WGS84) | Interruption Start | Interruption End |
|---------|-----------|----------|---------------|--------------|--------------------|------------------|
| 2 | 17 April 2015 | 16 March 2016 | 108.761 | 27.85246 | | |
| 5 | 22 April 2015 | 16 March 2016 | 108.7325 | 27.88133 | 7 September 2015 | 23 October 2015 |
| 7 | 26 April 2015 | 16 March 2016 | 108.7217 | 27.8868 | | |
| 19 | 28 April 2015 | 15 March 2016 | 108.6994 | 27.91098 | 23 June 2015 | 6 July 2015 |
| 20 | 28 April 2015 | 15 March 2016 | 108.7029 | 27.90261 | 30 June 2015 | 23 October 2015 |
| 21 | 28 April 2015 | 15 March 2016 | 108.6998 | 27.90731 | 7 September 2015 | 23 October 2015 |
| 22 | 29 April 2015 | 15 March 2016 | 108.7075 | 27.90061 | | |
| 23 | 29 April 2015 | 15 March 2016 | 108.712 | 27.89908 | 7 September 2015 | 23 October 2015 |
| 24 | 29 April 2015 | 15 March 2016 | 108.7245 | 27.89716 | | |
| 27 | 2 May 2015 | 18 March 2016 | 108.7736 | 27.85997 | | |
| 28 | 2 May 2015 | 18 March 2016 | 108.7725 | 27.85966 | | |
| 29 | 4 May 2015 | 18 March 2016 | 108.7331 | 27.90562 | 1 September 2015 | 8 September 2015 |
| 30 | 4 May 2015 | 18 March 2016 | 108.7302 | 27.90692 | | |
| 31 | 14 May 2015 | 15 March 2016 | 108.697 | 27.78216 | | |
| 32 | 14 May 2015 | 15 March 2016 | 108.7005 | 27.78644 | 26 June 2015 | 30 October 2015 |
| 34 | 19 May 2015 | 8 April 2016 | 108.641 | 27.81311 | | |
| 35 | 20 May 2015 | 7 August 2016 | 108.6495 | 27.76345 | | |
| 36 | 20 May 2015 | 7 August 2016 | 108.6499 | 27.77022 | | |
| 37 | 20 May 2015 | 9 April 2016 | 108.6522 | 27.77409 | 14 November 2015 | 20 November 2015 |
| 38 | 21 May 2015 | 2 August 2016 | 108.6257 | 27.88032 | | |
| 39 | 22 May 2015 | 5 August 2016 | 108.6357 | 27.87258 | 16 September 2015 | 9 November 2015 |
| 40 | 22 May 2015 | 5 August 2016 | 108.6422 | 27.87614 | | |

**Table A2.** *Cont.*

| Plot ID | Start Date | End Date | Long. (WGS84) | Lat. (WGS84) | Interruption Start | Interruption End |
|---|---|---|---|---|---|---|
| 41 | 28 May 2015 | 30 June 2016 | 108.6579 | 27.91372 | | |
| 42 | 9 November 2015 | 2 August 2016 | 108.6471 | 27.91849 | | |
| 43 | 31 May 2015 | 1 July 2016 | 108.6558 | 27.92395 | | |
| 44 | 2 June 2015 | 9 April 2016 | 108.7692 | 27.97725 | | |
| 45 | 2 June 2015 | 28 April 2016 | 108.755 | 27.97899 | | |
| 46 | 2 June 2015 | 19 April 2016 | 108.7478 | 27.97599 | 1 July 2015 | 27 October 2015 |
| 47 | 4 June 2015 | 15 August 2016 | 108.7607 | 27.98081 | | |
| 48 | 5 June 2015 | 15 August 2016 | 108.7549 | 27.98858 | | |
| 49 | 13 June 2015 | 30 July 2016 | 108.7384 | 28.004 | | |
| 50 | 13 June 2015 | 30 July 2016 | 108.7409 | 28.00193 | | |
| 54 | 23 June 2015 | 28 July 2016 | 108.6826 | 27.91677 | | |
| 55 | 23 June 2015 | 28 July 2016 | 108.6838 | 27.91513 | 28 October 2015 | 4 November 2015 |
| 57 | 24 June 2015 | 16 November 2015 | 108.6862 | 27.78472 | 3 July 2015 | 13 November 2015 |
| 58 | 10 July 2015 | 8 August 2016 | 108.6769 | 27.93495 | 24 November 2015 | 17 February 2016 |
| 59 | 10 July 2015 | 14 September 2015 | 108.6747 | 27.93743 | | |
| 60 | 10 July 2015 | 1 April 2016 | 108.6676 | 27.95285 | | |
| 10 | 25 October 2015 | 30 July 2016 | 108.7412 | 28.00497 | | |
| 8 | 4 December 2015 | 13 August 2016 | 108.7705 | 27.97893 | | |
| 9 | 2 November 2015 | 27 July 2016 | 108.7394 | 27.90263 | | |
| 11 | 4 November 2015 | 27 July 2016 | 108.7733 | 27.85921 | | |
| 12 | 19 March 2016 | 27 July 2016 | 108.79 | 27.90854 | | |
| 13 | 13 November 2015 | 27 July 2016 | 108.7959 | 27.90966 | | |
| 14 | 8 April 2016 | 5 August 2016 | 108.6466 | 27.81613 | | |
| 77 | 18 March 2016 | 12 July 2016 | 108.7488 | 27.89871 | | |
| 76 | 18 March 2016 | 27 July 2016 | 108.7764 | 27.85929 | | |
| 75 | 19 March 2016 | 10 August 2016 | 108.7725 | 27.99097 | | |
| 15 | 19 March 2016 | 18 July 2016 | 108.7708 | 27.98718 | | |
| 74 | 22 March 2016 | 24 July 2016 | 108.7411 | 27.83366 | | |
| 73 | 22 March 2016 | 12 May 2016 | 108.7505 | 27.82971 | | |
| 72 | 23 March 2016 | 10 August 2016 | 108.7774 | 27.98696 | | |
| 71 | 23 March 2016 | 12 August 2016 | 108.7817 | 27.99012 | | |
| 70 | 25 March 2016 | 14 August 2016 | 108.7466 | 27.9684 | | |
| 68 | 13 April 2016 | 15 August 2016 | 108.7664 | 27.99333 | | |
| 67 | 27 March 2016 | 20 July 2016 | 108.781 | 28.00514 | | |
| 66 | 27 March 2016 | 11 August 2016 | 108.7815 | 27.99568 | | |
| 65 | 28 March 2016 | 11 August 2016 | 108.7746 | 27.99835 | | |
| 64 | 28 March 2016 | 24 June 2016 | 108.77 | 28.0013 | | |
| 63 | 29 March 2016 | 12 August 2016 | 108.7759 | 27.96883 | | |
| 62 | 29 March 2016 | 11 August 2016 | 108.7852 | 27.98265 | | |
| 61 | 30 March 2016 | 13 July 2016 | 108.7564 | 28.02354 | | |
| 53 | 2 April 2016 | 23 April 2016 | 108.6678 | 27.97648 | | |
| 51 | 4 April 2016 | 10 July 2016 | 108.7497 | 28.02282 | | |
| 18 | 4 April 2016 | 30 July 2016 | 108.7404 | 28.02277 | | |
| 16 | 6 April 2016 | 31 July 2016 | 108.5902 | 27.91872 | | |
| 17 | 6 April 2016 | 25 June 2016 | 108.6088 | 27.92623 | | |
| 78 | 10 April 2016 | 28 July 2016 | 108.69 | 27.90617 | | |
| 79 | 10 April 2016 | 28 July 2016 | 108.6875 | 27.89934 | | |

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
