# Peer review of "Synthesizing Remote Sensing and Biophysical Measures to Evaluate Human–wildlife Conflicts: The Case of Wild Boar Crop Raiding in Rural China"

_remotesensing, doi:10.3390/rs12040618_

Round 1

Reviewer 1 Report

This study used a combination of game cameras, Landsat imagery, and household surveys to evaluate wild boar damage in China. It is an interesting study that will be of interest to readers and managers. Overall, I found the manuscript to be well-written with a few exceptions as will be explained below. The study design and analyses seem fine. Overall, great job. Line numbers below reference particular issues I had with the paper.

I found the abstract to be concise and well written. Introduction also was well written, nice job! Capitalize the S in Sus. …….populations, and, or mitigate…..

122-140. How were areas for camera trap deployment decided? How does the June to August period correlate with damage (ie. Is this the critical time)?

These numbers seem really low. However, glad you are indicating this is a proxy for the population density. But why present results in methods? Reword or spell out 156. Do not start a sentence with digits. What was the reason for wanting 650 useable surveys? WDVRI was already defined.

169-185. This is the only area of the manuscript that I felt had issues with the writing. The methods should just stick to the methods. Ecology of the species would be best in the introduction or in a section on species ecology. This section should just focus on the methods used. This would make this section easier to comprehend.  

242-245. A few more details here would be useful to see. What was the maximum number of Sus scrofa seen in a day at a camera? How many camera had no boars?

260-262. I am having trouble interpreting this section. What do the numbers in Table 2 mean? If you are truly showing change in population since 1990 why are 1990 data negative? Are the 2015-2016 date the baseline? I would think 1990 would be the base. I think a better explanation in the table caption and the text would easily clarify the confusion.

325-327. Are they really? The data do not show any indication of how strong the relationships are. AIC only shows you the best models. How much actual variation are they explaining? Where are the moderately high R2 you reference? The only R2 I see are for radiometric rectification and that does not address this issue.

I like the “telescope” concept, nice.

373-376. But without data on amount of damage I don’t think you actually show this. Your densities were so low it’s difficult to interpret. Is one boar every 30, 50, or 100 days enough to wipe out significant amounts of crops?

385-390. Yes, agree!

Good luck with revisions.

Author Response

Thank you for your attention; please see the attachment.

Reviewer 2 Report

The authors analysed how the density of wild boar affect crops in China over time, and which was the impression gathered by the farmers. For this, wild boar locations were obtained with camera-traps. Wild boar density was calculated and related by linear regression with several vegetation types and other variables. Variables were obtained from several remote sensing sources. Regressions were projected to several years. How wild boar density changed over time was analysed with GWR.
The paper is very interesting and is very well written, although the methodology is hard to follow. Some parts need to be explained with more detailed. In fact, there are some decisions I did not understand. Some results are missing. The discussion is too brief. It does not discuss alternatives to the methodology. In fact, the discussion cites very few references.
1. What do you mean with anniversary Landsat?
2. Why did the authors analyse only four dates? Please, justify the selection of dates.
3. Present the formula of WDRVl.
4. Explain the k parameter in IDW. Please, justify the selection of k values.
5. I do not understand why were Thiesen polygons calculated? Why not using the IDW raster? It seems to me that the methodology was complicated unnecessarily. A flow-chart figure explaining the whole methodological process may be very useful.
6. Why using clustered random samples and not independent samples? Biases may be introduced in the model. In fact, this is recognised in the discussion. I would use only one point for each random sample.
7. Please, justify the use of OLS. Why are you supposing that the relationships between dependent and independent variables are linear? GAM may provide better results. Why not using GWR always? Using always GWR would simplify the methodology and put all the analyses in a spatial context.
8. Present a figure of the study area. This is essential to get a better understanding of the manuscript.
9. Present a figure with the distribution of camera-traps. If not, it is not possible to understand if there are spatial biases in the distribution of the camera-traps, as recognised in the discussion.
10. Present-a figure with the IDW interpolation.
11. Present a figure with the Thiesen polygons.

Specific comments:
L 59: correct to Sus scrofa.
L 143: What are the 1298 points?
L 157: Please, explain the Likert scale. Provide a reference.
L 243-249: These results can be presented as a table. The text here is hard to follow.
L 249-250: Present these results as a graph. This will simplify the text.
L 258-292: Do not repeat results that are presented in the tables.

Author Response

(The authors gave the same response as above.)

Reviewer 3 Report

Wild boars population has big influence on damage of field cereal production. each country has different approach to deal with this problem. Remote sensing of wild boars activity can have positive influence on general control of this problem.

Author Response

Thank you, Reviewer #3, for your attention to our manuscript.

Reviewer 4 Report

Very interesting paper, and I found your approach to be unique and challenging. Some maps would be very helpful to show a) the distribution of vegetation types in your study area b) the distribution of villages and surveyed areas and c) the modeled density of boars used to compare to the changes in remotely sensed variables. Without any of this information, I have some serious concerns:

I firmly believe that relying on a single year of boar observations is too limiting. Populations vary substantially, and habitat use can vary substantially from one year to another.  There is no mention of how different weather may have contributed to vegetation differences, and thus habitat use by the boars. Especially since you are hindcasting density, it seems like you could at least look to overall climate as another set of variables. Even with that, I seriously question the validity of a single season worth of observational data to infer back 25 years.  I need further explanation on how the k-values relate to specific hypotheses on boar density and how well the camera traps are capturing that density.  Simply stating that the k1 (furthest camera weighted highest) because of statistical strength is not sufficient.  Showing where the social survey responses were in relation to habitat type would definitely help understand these results a lot more. 

Otherwise, I do think this has some merit. I just feel like you are extrapolating too far beyond the data. 

Author Response

(The authors gave the same response as above.)

Round 2

Reviewer 2 Report

I appreciate the effort made by the authors to clarify my doubts, but unfortunately I continue to have some concerns about the methodology.   Some of my previous doubts remain: 1) I continue to do not understand what the authors mean with anniversary dates. 2) Why the images were selected at intervals of five years? The authors selected five dates: 1990, 1996, 2002, 2011, 2016. Some biological reasons must be provided to explain the interval. 3) Please, provide more details the Likert scale. The reader will not understand what is it. 4) I do not understand why clustering sampling saves time. And the problem with biases is not resolved.   The authors related wild boar density with the following variables, first with OLS and then with GWR: WDRVI, elevation, TRMI, slope, and aspect. If I understood correctly, vegetation is not included directly in the models. Why not including vegetation as a factor? This must explained in detail. Further, why estimating vegetation cover by Thiessen polygons? Why not classifying the images? If the study aims to analyse changes over time, but the only temporal variable analysed by GWR is WDRVI. This will provide more information to the temporal analysis.   The authors represented the study area in Figure 1. Do you have cameras outside the study area? So, which is the study area used for analyses? Please, increase the contrast of the Landsat image. All the study area appears too dark.   Figure and table captions must provide more information. For example, Table 1 do not indicate what is TRMI. This must be explained in the caption.

Author Response

Thank you for your continued contribution; please see the attachment.
